# New Therapeutic Interventions for Kidney Carcinoma: Looking to the Future

**DOI:** 10.3390/cancers14153616

**Published:** 2022-07-25

**Authors:** Lucio Dell’Atti, Nicoletta Bianchi, Gianluca Aguiari

**Affiliations:** 1Division of Urology, Ospedali Riuniti University Hospital, 60126 Ancona, Italy; lucio.dellatti@ospedaliriuniti.marche.it; 2Department of Medical Sciences, University of Ferrara, 44121 Ferrara, Italy; nicoletta.bianchi@unife.it; 3Department of Neuroscience and Rehabilitation, University of Ferrara, 44121 Ferrara, Italy

**Keywords:** kidney cancer, signaling, autophagy, p53, drugs

## Abstract

**Simple Summary:**

Renal cell carcinoma (RCC) in metastatic form is a lethal pathology difficult to treat; therefore, the research of new therapeutic options for the treatment of metastatic patients is crucial to improve quality of life and overall survival. Recently, new signaling pathways and biological processes involved in cancer development and progression by scientific research community have been identified. These components including factors affecting angiogenesis, cell migration and invasion, autophagy and ferroptosis that are dysregulated in kidney cancer represent novel possible target molecules. In this work, we discuss current and new therapies for kidney cancer treatment; in particular, agents targeting new molecules involved in renal carcinogenesis that in future might become more powerful drugs for the cure of metastatic RCC.

**Abstract:**

Patients suffering from metastatic renal cell carcinoma (mRCC) show an overall survival rate of lower than 10% after 5 years from diagnosis. Currently, the first-line treatment for mRCC patients is based on antiangiogenic drugs that are able to inhibit tyrosine kinase receptors (TKI) in combination with immuno-oncology (IO) therapy or IO-IO treatments. Second-line therapy involves the use of other TKIs, immunotherapeutic drugs, and mTOR inhibitors. Nevertheless, many patients treated with mTOR and TK inhibitors acquire drug resistance, making the therapy ineffective. Therefore, the research of new therapeutic targets is crucial for improving the overall survival and quality of life of mRCC patients. The investigation of the molecular basis of RCC, especially in clear cell renal cell carcinoma (ccRCC), has led to the identification of different signaling pathways that are involved in renal carcinogenesis. Most of ccRCCs are associated with mutation in *VHL* gene, which mediates the degradation of hypoxia-inducible factors (HIFs), that, in turn, regulate the pathways related to tumorigenesis, including angiogenesis and invasion. Renal tumorigenesis is also associated with the activation of tyrosine kinases that modulate the PI3K-Akt-mTOR pathway, promoting cell proliferation and survival. In ccRCC, the abnormal activity of mTOR activates the MDM2 protein, which leads to the degradation of tumor suppressor p53 via proteasome machinery. In addition, p53 may be degraded by autophagy in a mechanism involving the enzyme transglutaminase 2 (TG2). Suppression of wild-type p53 promotes cell growth, invasion, and drug resistance. Finally, the activation of ferroptosis appears to inhibit cancer progression in RCC. In conclusion, these pathways might represent new therapeutic targets for mRCC.

## 1. Introduction

Metastatic renal cell carcinoma (mRCC) is a lethal form of RCC that is very hard to cure. In fact, subjects with mRCC at 5 years from diagnosis show an overall survival (OS) lower than 10% [1]. Approximately 30% of RCC patients develop metastatic disease at diagnosis, and disease recurrence occurs in about 30% of patients after surgical treatment [2]. The detection of altered signals involved in the pathogenesis of this disease has allowed for the development of specific drugs. In particular, sunitinib and sorafenib (VEGFR inhibitors), as well as other tyrosine kinase inhibitors (TKI), such as pazopanib, axitinib and cabozantinib, have been approved by the FDA for the first-line treatment of mRCC [3]. Temsirolimus and everolimus, inhibitors of the mammalian target of rapamycin (mTOR), were recommended in the first- and second-line treatment of mRCC, respectively [3,4]. The treatment using these drugs has shown improvements in progression-free survival (PFS) after phase III clinical trials compared with interferon-alpha or other drugs [3,4]. However, no advantages in survival rate were seen with these agents due to treatment toxicity and drug resistance [2,5]. Moreover, patients who received everolimus in second-line treatment had significantly inferior outcomes compared with other treatments; therefore, the current guidelines have not recommended the use of mTOR inhibitors alone after anti-angiogenic therapy [4]. Currently, combined therapies with TKIs and immune checkpoint inhibitors (ICI) or combinations of ICIs are used for the first-line treatment of RCC [6]. However, the research and testing of more efficient drugs are crucial for improving the treatment of mRCC.

Here we summarize the clinical aspects, the main/novel pathways involved in renal carcinogenesis, and possible therapeutic targets useful for the generation of new drugs.

## 2. Clinical Aspects

Worldwide, renal cell carcinoma (RCC) is the sixth most frequently diagnosed cancer in men and the tenth in women, accounting for 5% and 3% of all oncological diagnoses, respectively [7]. RCC is mainly divided into three well-defined histotypes; clear cell renal cell carcinoma (ccRCC) makes up about 70% of all kidney cancers. Papillary RCC (pRCC) is the second most common renal tumor. Two subtypes of papillary renal cell cancer have been recognized: papillary type 1 and papillary type 2. The third most common RCC is chromophobe (chRCC) which has a largely empty cytoplasm and low mitotic rate. In general, this tumor has the lowest risk of developing metastases [8,9].

### 2.1. Tumor Staging

The therapeutic options and management of RCC are stage-dependent; consequently, accurate staging is essential to effective management [9]. Kidney primary tumors can be classified in the following groups:

*T1:* Tumor is confined to the kidney and is subdivided into T1a and T1b. T1a: tumor is 4 cm or less in size; T1b: tumor is between 4 and 7 cm. 

*T2:* Tumor is limited to the organ and is divided into T2a and T2b. T2a: tumor is between 7 and 10 cm; T2b: tumor is more than 10 cm. 

*T3:* Tumor extends into major veins or perinephric tissues and is divided into T3a, T3b, and T3c. T3a: tumor extends into the renal vein or invades the pelvicalyceal system or spreads into perirenal and/or renal sinus fat; T3b: tumor extends into vena cava below diaphragm; T3c: tumor extends into the vena cava above the diaphragm or invades the wall of the vena cava. 

*T4:* Tumor invades beyond Gerota fascia (including contiguous extension into the ipsilateral adrenal gland). 

Renal cancer staging is based on the tumor node metastasis (TNM) classification system and includes four different stages.

*Stage I:* Includes T1 tumors without lymph node or distance metastasis. Partial nephrectomy (PN) or radical nephrectomy (RN) are recommended. 

*Stage II:* Includes T2 tumors without lymph node or distance metastasis. Surgery including PN and RN is an option for the resection of tumor masses.

*Stage III:* Includes T3 tumors without lymph node or distance metastasis and T1–T3 tumors with metastasis in regional lymph nodes, but not distance metastasis. RN or PN (if clinically indicated) are recommended as surgical treatment.

*Stage IV:* Includes T4 tumors with any lymph node without distance metastasis as well as T1–T4 tumors with any lymph node and distance metastasis. RN could be an option for patients with localized disease. Cytoreductive nephrectomy (CN) before systemic therapy might be an option for mRCC patients with surgically treatable primary tumors. In fact, studies conducted in the cytokine era showed that combined treatment with CN and interferon enhanced the survival of patients compared with subjects treated with interferon alone. Conversely, the CARMENA study reported that OS in patients treated with sunitinib alone was not inferior to CN followed by sunitinib [10,11]. Based on these observations, the immediate systemic treatment for mRCC patients is recommended [10].

### 2.2. Trends in the Surgical Management of RCC

RN remains the gold standard treatment for localized renal masses in any patient not suitable for nephron-sparing surgery (NSS). In particular, the laparoscopic technique shows equivalent oncologic outcomes as compared to open RN (ORN), but with a faster recovery, lower blood loss, and decreased complication rates [12]. Laparoscopy quickly became the favorite approach for patients with localized renal cancer requiring RN [13,14]. The emergence of the robotic platform in the early 2000s has again shifted the surgical approach to this disease. This technique has also affected RN, as seen by the increasing utilization of the robotic platform. The Premier Healthcare database shows that the use of the robotic approach increased from 1.5% in 2003 to 27% in 2015 [15]. However, RN with the robotic approach shows no clear benefit as compared to a laparoscopic one but is significantly easier to learn and perform as compared to the laparoscopic approach [16]. Recently, the rise of the robotic platform has significantly changed the NSS approach. Outcome data for robotic partial nephrectomy (RPN) compared to laparoscopic (LPN) and open (OPN) have been promising, showing RPN to be superior to the other techniques. Different studies have shown that RPN is better than OPN for blood loss, transfusions, complications, hospital stay, readmissions, overall mortality, and the recurrence rate of disease [17]. The rapid evolution of the robotic-assisted approach has become the most common and favorable surgical approach for the management of RCC.

### 2.3. Management of RCC Patients after Diagnosis

Nonmetastatic RCC patients with localized disease and subjected to partial or radical nephrectomy are monitored by surveillance. Follow-up should be personalized based on the patient’s needs. Adjuvant therapy, by using TKIs for high-risk nonmetastatic RCC patients after nephrectomy, was approved by the FDA, but is not recommended by EAU guidelines. In fact, the S-TRAC study showed that treatment with sunitinib improved disease-free survival (DFS), yet without any OS benefit [10]. Recently, the availability of novel drugs, including ICIs, has improved the efficacy of adjuvant therapies. In this regard, the Keynote-564 trial reported that treatment with pembrolizumab enhanced DFS in high-risk nonmetastatic RCC patients subjected to nephrectomy as compared to placebo [10,18]. These observations indicate that pembrolizumab could be used as adjuvant therapy for high-risk RCC. However, before recommending this therapy, it would be appropriate to know the data on OS, currently not yet available [10].

Metastatic RCC patients are risk-stratified into favorable, intermediate, and poor-risk categories. Recently, the integration of molecular data with annotated genomic models showed an improved stratification of patients across the different risk groups. In particular, the standard of care for the treatment of favorable risk patients is the use of pembrolizumab/Axitinib, pembrolizumab/Lenvatinib, or nivolumab/cabozantinib. For intermediate and poor-risk, the recommended treatment is nivolumab/cabozantinib, pembrolizumab/axitinib, pembrolizumab/Lenvatinib, or nivolumab/ipilimumab [6].

## 3. Signaling Related to RCC

The most common RCC (ccRCC histotype) is linked to well-known altered-signaling pathways such as the von Hippel–Lindau (VHL), vascular endothelial growth factor receptor (VEGFR), and the phosphoinisitide-3 kinase/mammalian target of rapamycin (PI3K/mTOR) protein kinases. Currently, it has emerged that new pathways, including receptor tyrosine kinases (RTKs) hyper-activation, p53 damage, and autophagy dysfunction, are involved in ccRCC pathogenesis and drug resistance.

The main biological processes involved in renal carcinogenesis are described below and in Figure 1.

### 3.1. VHL-HIF-VEGFR-mTOR

VHL loss of function is a common event in ccRCC, and leads to the abnormal activation of well-known signaling pathways, including hypoxia-inducible factors (HIFs), VEGFR, and mTOR [19,20]. The dysfunction of VHL protein prevents the ubiquitination and degradation of HIF factors via proteasome machinery. Hypoxia conditions and VHL inactivation lead to the expression of VEGF, promoting angiogenesis and tumor growth [21]. VHL loss of function also causes the hyperactivation of mTOR, which correlates with tumor progression and poor outcomes in ccRCC patients [21]. In addition, HIF-1α mediates the expression of TWIST-related protein 1 (TWIST1) and sustains invasion and metastasis through the regulation of Snail/Slug/ZEB1 axis. The activation of this pathway promotes the epithelial to mesenchymal transition (EMT) by the downregulation of E-cadherin, which, in turn, correlates with the loss of the epithelial and the acquisition of the mesenchymal phenotype [22,23]. This signal is further associated with inflammation factors, such as transforming growth factor beta (TGF-β), which stimulates the secretion of cytokines, promoting drug resistance and invasion [24].

### 3.2. RTK-PI3K-Akt

The increased expression and activation of VEGFR due to the accumulation of HIF is a well-validated pathogenic mechanism in ccRCC [25]. However, other tyrosine kinases, including platelet-derived growth factor receptor (PDGFR), tyrosine-protein kinase receptor UFO (Axl), and mesenchymal epithelial transition receptor (MET), are involved in ccRCC biology [25]. The binding of VEGFR and PDGFR with their ligands VEGF and PDGF, respectively, activates the PI3K, which, through the generation of phosphatidylinositol-3,4,5-triphosphate (PIP3), enhances AKT kinase activity. The activation of AKT inhibits apoptosis and stimulates tumor progression, inactivating pro-apoptotic proteins such as procaspase 9, BCL2, an associated agonist of cell death (BAD), and apoptosis signal-regulating kinase 1 (ASK1) in ccRCC cells. In addition, the abnormal activation of VEGFR and PDGFR signaling leads to the increased activity of mTOR through AKT phosphorylation and promotes protein synthesis and cell growth [19]. The dysregulation of MET, another tyrosine kinase receptor activated by the hepatocyte growth factor (HGF), induces the activation of both the PI3K-Akt and Ras/MAPK pathways which promote cell growth and metastasis in kidney cancer [26]. Moreover, MET is implicated in the mechanisms of resistance to targeted therapies, including EGFR and VEGFR inhibitors [27]. Finally, the overexpression of Axl, a tyrosine kinase receptor belonging to the TAM family, which is activated by the growth arrest-specific protein 6 (GAS6) in RCC, was observed. In kidney cancer, the activation of Axl by Gas6 or autocrine signaling leads to tumor cell growth, metastasis, invasion, EMT, angiogenesis, and drug resistance, mainly in a PI3K-Akt-mTOR dependent manner [28,29]. These observations also suggest that Axl tyrosine kinase is a potential therapeutic target in RCC.

### 3.3. p53 Related Signaling

Mutations of the tumor suppressor p53 in ccRCC are rare; however, it was observed that wild-type p53 is expressed at lower levels in tumor cells, suggesting that this protein might be suppressed by other mechanisms [30,31]. In this regard, the dysfunction of the *VHL* gene, mutated in most of kidney carcinomas, is associated with the reduction or absence of p53 expression. Moreover, functional VHL protein is able to promote p53 function, enhancing the expression of its downstream effectors p21 and Bax, which are involved in cell-cycle control and apoptosis [32]. The tumor suppressor p53 might also be removed in kidney cancer cells by the proteasome system. In fact, it was reported that the ubiquitin protein RBCK1 could function as an oncogene in RCC, leading to p53 ubiquitination and, consequently, to its degradation by proteasome machinery [33]. Consistently, it was found that in different ccRCC cell lines, the activation of mTOR promotes the expression of E3 ubiquitin-protein MDM2, which, in turn, induces p53 ubiquitination and degradation by proteasome [34]. Taken together, these findings suggest that in ccRCC, p53 might be inactivated by proteasome degradation in a mTOR-MDM2-dependent manner. Another way to suppress p53 in ccRCC is for it to be captured and degraded by autophagy. In fact, we have observed that in different ccRCC cell lines, autophagy is increased compared with non-tumor kidney cells and, in cancer cells, p53 is sequestered and degraded into autophagosomes [35]. Consistently, the inhibition of autophagy by *ATG7* gene silencing reduces cell proliferation, migration, and EMT, enhancing the expression of both p53 and p21 proteins [35]. Recently, it was reported that the levels of the enzyme transglutaminase 2 (TG2) correlated with those of microtubule-associated protein 1A/1B-light chain 3 (LC3), supporting the relationship between TG2 and the autophagic pathway [36]. Moreover, TG2 levels are also associated with EMT since this enzyme promotes the expression of Vimentin [37]. Importantly, TG2 is crucial for the translocation of p53 into autophagosomes through a linkage with TG2. In fact, TG2 interacting with p53 and the autophagic protein p62/SQSTM1 forms a heterotrimeric complex (p53-TG2-p62), which leads to p53 depletion in ccRCC cells [38]. Furthermore, the TG2-mediated decrease of p53 modulates HIF-1α by p300 and, consequently, the levels of VEGF affecting angiogenesis [39]. Taken together, these findings support the targeting of MDM2 and TG2 to develop new therapies for mRCC patients.

### 3.4. Ferroptosis

Another field of investigation is represented by the process of ferroptosis in the control of cell death [40]. Ferroptosis is a newly discovered form of iron-dependent oxidative cell death, characterized by the lethal accumulation of lipid-based reactive oxygen species (ROS) [41]. Ferroptosis is different from other cell deaths, including apoptosis, necrosis, and autophagy and is under the control of the Hippo-YAP/TAZ pathway [40]. Inhibitors of the cystine/glutamate exchange system (system xc^−^) such as erastin, sulfasalazine (SAS), or sorafanib, lead to the decrease of intracellular GSH, promoting the accumulation of ROS, which causes cell death by ferroptosis. The activation of ferroptosis also occurs by RAS-selective lethal small molecular-3 (RSL-3), an inhibitor of Glutathione Peroxidase 4 (GPx4). RSL-3 induces ferroptosis without decreasing GSH levels or inhibiting the system xc^-^, suggesting that this compound may activate ferroptosis by a different initiating mechanism [41]. This process in kidney carcinoma represents a novelty and depends on the amassment and density of the cancer cells. In fact, erastin- and RSL3-induced ferroptosis is inhibited by the enhanced cell density in RCC cells through a mechanism involving the inactivation of TAZ. This transcription factor is the predominant Hippo effector in RCC cells, and is phosphorylated, retained in the cytosol, and subjected to proteasomal degradation in high cell-density conditions [40]. TAZ inactivation prevents the expression of the epithelial membrane protein 1 (EMP1), which, in turn, promotes the upregulation of nicotinamide adenine dinucleotide phosphate (NADPH) Oxidase 4 (NOX4), a reactive oxygen species (ROS)-generating enzyme essential for ferroptosis [40]. Taken together, these observations indicate that high cell-density inhibits cell death by ferroptosis in kidney cancer cells through the inactivation of the TAZ/EMP1/NOX4 pathway. Because erastin administration reduced tumor growth in 786-O xenograft models of RCC, the induction of ferroptosis might represent a therapeutic target for patients with advanced kidney carcinoma.

## 4. Current and New Targeted Therapies for RCC Treatment

The pharmacological therapies for the treatment of mRCC patients move fast. In fact, the monotherapy with tyrosine kinase inhibitors targeting the VEGF receptor to inhibit angiogenesis was replaced with more effective therapies.

Here, we explore the current and novel possible targeted drugs for the treatment of mRCC.

### 4.1. TK Inhibitors

Treatment with first-generation TKIs such as sunitinib, sorafenib, and pazopanib in mRCC patients leads to the development of primary and acquired resistance to these drugs. Thus, these therapeutic agents could be replaced with more efficient TKIs, such as cabozantinib, axitinib, Lenvatinib, and tivozanib [42,43]. Cabozantinib is an inhibitor of multiple tyrosine kinases, including VEGFR, MET, and Axl, which are associated with aggressive disease and poor survival [42]. The treatment with cabozantinib showed an increased PFS in mRCC patients compared with sunitinib [44]. Axitinib is an anti-angiogenic multi-receptor inhibitor that, in clinical trials, showed greater objective response rates and improved median PFS compared with sorafenib [4,43]. Lenvatinib is a multitarget tyrosine kinase inhibitor that inhibits VEGFR, FGFR, PDGFR, RET, and KIT. The administration of this drug has shown antitumor properties against mRCC [45]. Finally, tivozanib, a novel selective VEGFR inhibitor, was approved for the treatment of advanced RCC by the European Medicines Agency (EMA) [43,46]. The treatment with tivozanib showed a PFS advantage compared with sorafenib; however, overall survival results favored sorafenib, yet other studies should be carried out to evaluate the efficacy of this drug [46,47]. The combination of axitinib, cabozantinib, and lenvatinib with different PD-1 immune checkpoint inhibitors in a first-line setting has shown superior efficacy in patients with advanced RCC compared with single-drug therapies [6,48]. Therefore, the latest guidelines recommend the use of these TKIs in first-line treatment in combination with immunotherapeutic drugs [6].

### 4.2. HIF2 Alpha Antagonists and PI3K-Related Inhibitors

The targeting of the HIF pathway seems a promising option for treating patients with metastatic ccRCC. In fact, treatment with the HIF2 antagonist PT2399 showed greater activity than sunitinib and was well tolerated in mice models for ccRCC [49]. However, prolonged PT2399 treatment leads to resistance. Moreover, some cases of drug resistance were observed despite HIF2 inactivation, suggesting that some ccRCCs are HIF2-independent [49]. The same research group has also developed and tested a second HIF2 antagonist (PT2385) that is able to inhibit HIF2 binding as well as HIF-2-related gene expression. PT2385 was safe and active in a first-in-human phase I clinical trial of patients with extensively pre-treated ccRCC; however, follow-up data on PFS and OS are not yet available. As with PT2399, the prolonged treatment with PT2385 leads to drug resistance, likely due to HIF2 mutations [50]. Another HIF2 inhibitor, belzutifan, is an oral small molecule used for the treatment of solid tumors, including renal cell carcinoma. In a phase-1 study, belzutifan was well tolerated and demonstrated preliminary anti-tumor activity in patients pre-treated with antiangiogenic drugs. The main side effect in patients treated with belzutifan is anemia due to erythropoietin reduction [51]. Recently, a phase-II clinical trial using belzutifan in patients with renal carcinomas associated with VHL disease was completed. Results indicate that the treatment with belzutifan induces a reduction in tumor size in most patients [52]. In particular, about half of enrolled patients (49%) treated with belzutifan have shown an objective response. Only a few patients (3%) had progressive disease, and one subject (2%) discontinued the treatment because of an adverse event [52]. The efficacy of belzutifan, combined with these modest side effects, make this drug an important option for mRCC treatment.

The abnormal activation of PI3K is involved in the development of renal cell carcinoma, making this kinase and its relative signaling an attractive target for therapeutic intervention. Phase-II studies of mRCC patients previously treated with anti-VEGFR drugs by using MK-2206, an allosteric inhibitor of AKT, did not show significant differences when compared to studies of patients treated with the mTOR inhibitor everolimus [53]. In addition, a significant number of patients treated with MK-2206 developed a rash and hyperglycemia, as observed in patients treated with other PI3K-pathway inhibitors, such as BEZ235, apitolisib, and buparlisib [53]. Treatments with BEZ235 or apitolisib, dual PI3K and mTOR inhibitors in mRCC patients, showed high toxicity without objective improvements [54,55]. Finally, the treatment with buparlisib, a pan-class I phosphoinositide 3-kinase inhibitor, in combination with bevacizumab, showed different side effects, including elevated lipase/amylase, anorexia, and psychiatric disorders. Moreover, the objective responses obtained by this combined therapy were lower compared with the combination of bevacizumab and mTOR inhibitors [56].

The modest efficacy of these agents observed in RCC patients might be explained by the extensive crosstalk and negative feedback mechanisms typical of PI3K-related pathways [53].

### 4.3. Anti MDM2 Drugs and Proteasome Inhibitors

Since p53 may be removed by MDM2-proteasome machinery in kidney tumor cells [34], this system represents a possible target for cancer therapy. In this regard, the inhibition of MDM2 reduced tumor size in pre-clinical models for ccRCC. Furthermore, the combined treatment of MDM2 inhibitors and everolimus showed a synergistic effect decreasing cell growth in ccRCC for “in vitro” and “in vivo” models [57]. However, specific clinical trials targeting MDM2 in kidney cancer have to still be planned. Conversely, different clinical trials in solid tumors and hematologic neoplasms by using MDM2 inhibitors, such as RG7112, idasanutlin, APG-115, and ALRN-6924, have already completed phase-1 trials [58,59]. Most of these seem well tolerated except for RG7112, which shows different side effects. Nevertheless, some MDM2 inhibitors show ineffectiveness, while clinical trials testing other MDM2 antagonists are currently in progress [58].

The targeting of proteasome machinery with specific inhibitors might be another way to preserve wild-type p53 and arrest tumor growth. The use of bortezomib, a small inhibitor molecule of proteasome, is able to induce apoptosis in different RCC cell lines. Unfortunately, a phase-II study testing the combination of sorafenib and bortezomib did not show improvements in PFS compared with sorafenib monotherapy despite the combined drugs being well tolerated [60]. Another promising proteasome blocker is carfilzomib; this agent is a specific inhibitor for the chymotrypsin-like active site of the 20S proteasome. The treatment with this compound showed antitumoral activity not only in RCC tumor cell lines but also in a patient-derived xerograph (PDX) [61]. Nevertheless, in a phase-II study with mRCC patients it emerged that carfilzomib failed clinical trials since all enrolled patients had disease progression and different side effects [62]. Taken together, these observations suggest that proteasome inhibitors seem unsuitable for the treatment of metastatic renal carcinoma.

### 4.4. Autophagy and TG2 Inhibitors

Patients suffering from mRCC treated with multiple targeted therapies have acquired drug resistance and a loss of efficiency. Therapy resistance might be caused by drug sequestration in lysosomal vesicles. Actually, it was demonstrated that autophagy is able to remove and degrade the tyrosine kinase inhibitor sunitinib in kidney cancer cells [63]. In addition, we have reported that in kidney cancer cells, p53 is removed by the autophagic system, suggesting that the progression of kidney carcinoma could be associated with the activation of autophagy [35]. Therefore, the targeting of autophagy could be an appealing idea for the treatment of metastatic kidney cancer. It was observed that the modulation of autophagy might enhance the cytotoxicity of therapeutic agents and reduce drug resistance [64]. In particular, clinical trials using the autophagy inhibitor hydroxychloroquine (HCQ) in combination with everolimus have shown a better PFS compared with everolimus alone in mRCC patients previously treated with tyrosine kinase inhibitors. Moreover, the combined treatment was well tolerated since only in some cases drug-related toxicity was observed [65]. Other promising autophagy inhibitors, including ROC-325, are currently being used in pre-clinical studies and have shown anticancer activity [64]. Recent studies have further demonstrated that p53 sequestration into autophagosomes occurs through its interaction with the enzyme TG2, which connects p53 with the autophagic protein SQSTM1/p62 [38]. Moreover, autophagy-related drug resistance appears to be mediated by TG2; therefore, the inhibition of this enzyme might prevent drug resistance, improving cancer therapy [66]. Consistently, it was reported that treatment with streptonigrin, a TG2 inhibitor, was able to reduce cancer cell growth in a xenograft model of RCC, confirming that the targeting of TG2 exerts anticancer properties [66].

### 4.5. Ferroptosis Activators

Different studies have demonstrated that the induction of ferroptosis by erastin and sorafenib is able to inhibit tumor progression and retrieve therapeutic effectiveness. Therefore, the combination of chemotherapeutic drugs and ferroptosis inducers could be a further option to improve the overall survival of patients with advanced kidney carcinoma [41]. Recently, it was reported that treatment with artesunate (ART), a traditional Chinese medicine drug, has shown anticancer properties. In fact, ART is able to reduce cell proliferation and kill tumor cells in different RCC cell lines. In particular, ART works by increasing the number of ROS species and inhibiting GPX4 in a mechanism involving p53, ultimately leading to cell death by ferroptosis in KTCTL-26 cells. However, the mechanism of cell killing induced by ART in other RCC cell lines, such as Caki-1, 786-O, and A-498, is different since increased ferroptosis activity in these cells was not observed [67]. Taken together, these observations indicate that ART might act in different ways depending on tumor type or tumor heterogeneity. Another study has demonstrated that the induction of ferroptosis by erastin and RSL-3 in combination with everolimus inhibits the viability of RCC cells and may overcome drug resistance problems observed after everolimus treatment [68]. To evaluate the possible anticancer properties of ferroptosis in RCC, clinical trials using ferroptosis activators alone or in combination with other chemotherapeutic agents should be approached.

### 4.6. Immunotherapy

The continuous demand for more efficient therapies for the treatment of mRCC has led to the discovery of new agents, including immune-oncology drugs such as immune checkpoint inhibitors (ICIs). Immunotherapy for the treatment of advanced RCC by using interleukins or interferon has already been attempted, but with poor results [69]. Currently, new immune-based drugs such as nivolumab, ipilimumab, pembrolizumab, and avelumab, have been generated. Interestingly, it was observed that adjuvant treatment with pembrolizumab, an anti-programmed death 1 (PD-1) monoclonal antibody, prolongs DFS in high-risk patients with nonmetastatic RCC subjected to nephrectomy [18]. As previously mentioned, the treatment with ICIs combined with TKIs is recommended for the first-line treatment of mRCC [6]. In addition, dual treatment using the monoclonal antibody anti-PD ligand-1 (PD-L1), avelumab, and the TK inhibitor axitinib, was also approved by the FDA for the treatment of all risk groups of RCC [70]. Importantly, the combination of nivolumab (anti-PD-1 monoclonal antibody) and ipilimumab (anti-cytotoxic T-lymphocyte-associated protein-4 (CTLA4)) represents the standard of care for RCC patients under intermediate or poor-risk [6]. Other innovative immunotherapies include the use of modified cytokines, cellular therapies, and anticancer vaccines [71]. Despite promising results obtained with ICI-based therapies, long-term treatments using ICIs in RCC patients may induce mechanisms of resistance, decreasing the clinical benefits [69]. Therefore, the research of predictive biomarkers for the therapeutic response in RCC patients treated with ICIs is needed.

## 5. Conclusions

The limited efficacy of therapeutic treatment in mRCC patients is mainly due to the development of drug resistance and severe side effects after first- and second-line treatments. New drugs have been developed, and some agents seem very promising since prolong PFS and are well tolerated, while others are ineffective or exhibit high toxicity. Many TKIs in combination with ICIs are currently used by all risk groups in the first-line treatment of RCC. Moreover, immuno-oncology therapies are also recommended for the intermediate and poor-risk groups in RCC. Other inhibitors have shown interesting anticancer properties; in particular, the HIF2 inhibitor, belzutifan, used alone or in combination with other anticancer drugs, seems to improve PFS compared with conventional therapies. In addition, the administration of the autophagy inhibitor hydroxychloroquine, in combination with everolimus, enhances PFS as compared to a single treatment using everolimus in patients suffering from advanced RCC. These agents could represent new therapeutic options for the future treatment of RCC. Conversely, the use of Akt, PI3K, and proteasome inhibitors in mRCC patients does not reduce cancer progression and causes severe side effects. MDM2 and TG2 inhibitors, as well as ferroptosis activators, are able to inhibit tumor cell growth in pre-clinical models of RCC, but future clinical studies are needed to validate their anticancer activities. Furthermore, treatment using autophagy inhibitors and/or ferroptosis activators could limit drug resistance effects, re-sensitizing cancer cells to conventional therapy.

Overall, the research and validation of new targeting drugs, combined with the physiopathological features of kidney cancer, could lead to personalized medicine, improving the overall survival and quality of life of mRCC patients.

## Figures and Tables

**Figure 1 cancers-14-03616-f001:**
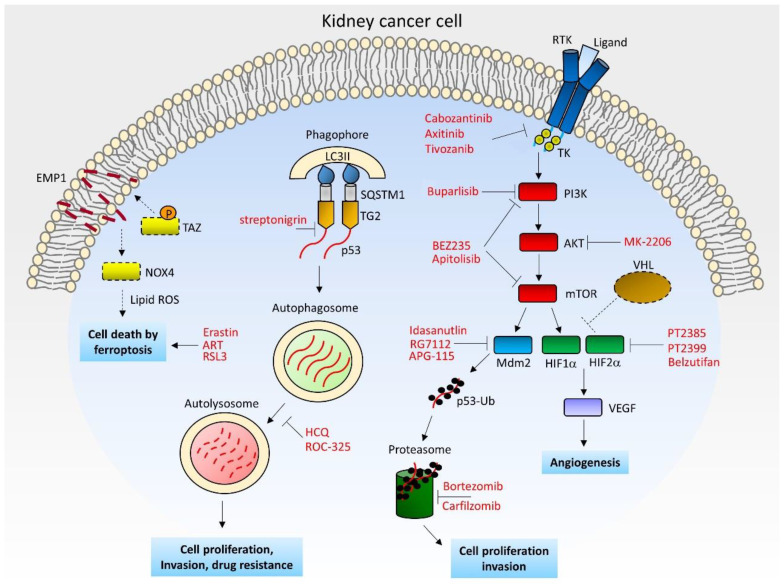
Schematic model of kidney cancer cell with the mainly pathogenic pathways/processes and relative inhibitors/activators. Dashed lines and boxes indicate the inactivated components in RCC. ART: Artesunate; EMP1: Epithelial membrane protein 1; HCQ: Hydroxychloroquine; HIF: Hypoxia-inducible factor; LC3II: Microtubule-associated protein light chain 3B; MDM2: Mouse double minute 2 homolog; mTOR: Mammalian target of rapamycin; NOX4: NADPH oxidase 4; PI3K: Phosphatidylinositol 3-kinase; ROS: Reactive oxygen species; RTK: Receptor tyrosine kinase; SQSTM1: Sequestosome-1; TAZ: Transcriptional co-activator with PDZ-binding motif; TG2: Transglutaminase 2; TK: Tyrosine kinase domain; Ub: Ubiquitin; VEGF: Vascular endothelial growth factor; VHL: von Hippel–Lindau.

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
