# Peer review of "New Therapeutic Interventions for Kidney Carcinoma: Looking to the Future"

_cancers, 2022, doi:10.3390/cancers14153616_

Round 1
Reviewer 1 Report
This is an interesting manuscript. However several points need to be clarified.
The data on current treatment for metastatic disease is not up do date and should incorporate new combinaisons ( TKI+IO or IO+IO). The role of nephrectomy for metastatic disease should be discussed based on the CARMENA data.
2.1 treatment of localized disease and staging are mixed up. Prognostic groups are for metastatic disease.
chapter 4: the sentence "The current therapies for the treatment of mRCC patients does not improve overall survival" is incorrect. For exampleTKI+IO or IO+IO combinaisons improve OS compared to sunitinib.
Current first line treatment for clear cell RCC is based on combinaisons and not on TKI alone. The paragraph 4.1 should be rewritten.
Precise numbers should be given concerning the efficacy of at least some of new drugs ( eg Belzutifan...)
Maybe there should be paragraph on new immunotherapies too.
The conclusion should be bullet point 5 and not 4. It should be partly rewritten as I don' t think we can consider axitinib or cabozantinib as new drugs
Author Response
Answers to Reviewer 1
This is an interesting manuscript. However, several points need to be clarified.
The data on current treatment for metastatic disease is not up do date and should incorporate new combinations (TKI+IO or IO+IO). The role of nephrectomy for metastatic disease should be discussed based on the CARMENA data.
In this regard, we have modified the Abstract inserting the following sentence:
“Currently, the first line treatment for mRCC patients is based on antiangiogenic drugs able to inhibit tyrosine kinase receptors (TKI) in combination with immuno-oncology (IO) therapy or IO-IO treatments”.
In addition, we have re-written the chapter 2.1. We have inserted a phrase based on the CARMENA data:
Stage 4: Tumor invades beyond Gerota fascia and adrenal gland. The surgical treatment for mRCC patients is debated. In fact, different studies suggest that nephrectomy may improve OS in mRCC subjects treated with targeted therapy. Conversely, the CARMENA study shows that OS and PFS in mRCC patients treated with Sunitinib alone was not inferior to nephrectomy followed by Sunitinib administration [11].
Moreover, we have inserted a new reference regarding the study CARMENA, reference number 11.
2.1 treatment of localized disease and staging are mixed up. Prognostic groups are for metastatic disease.
As mentioned above, the paragraph 2.1 has been deeply modified. In this paragraph, we would like to give brief information on the therapy based on the tumor stage. Other changes of this paragraph are the following:
Metastatic RCC patients are now treated with systemic targeted molecular therapies that have replaced immunotherapy (e.g. interferons-α). In particular, the standard care for the treatment of favourable risk patients is the use of Pembrolizumab/Axitinib, Pembrolizumab/Lenvatinib or Nivolumab/Cabozantinib. For intermediate and poor risk patients is recommended the treatment with Nivolumab/Cabozantinib, Pembrolizumab/Axitinib, Pembrolizumab/Lenvatinib or Nivolumab/Ipilimumab [6].
High risk nonmetastatic RCC patients subjected to nephrectomy might be treated with Pembrolizumab, because this treatment improves both disease free survival (DFS) and OS and showed lower toxicity compared with TKIs [12]. These findings support Pembrolizumab as new standard of care for nephrectomy RCC patients with high risk to develop metastasis [12].”
Chapter 4: the sentence "The current therapies for the treatment of mRCC patients does not improve overall survival" is incorrect. For example TKI+IO or IO+IO combinations improve OS compared to sunitinib.
The introduction to chapter 4 was modified. The new sentence is:
“The pharmacological therapies for the treatment of mRCC patients move fast. In fact, the monotherapy with tyrosine kinase inhibitors targeting the VEGF receptor to inhibit angiogenesis was replaced with more effective therapies.
Here, we explore current and novel possible targeted drugs for the treatment of mRCC.”
Current first line treatment for clear cell RCC is based on combinations and not on TKI alone. The paragraph 4.1 should be rewritten.
The paragraph 4.1 has been modified inserting new information as suggested by the Reviewer.
“4.1. TK inhibitors
The treatment with first generation TKI such as Sunitinib, Sorafenib and Pazopanib in mRCC patients leads to the development of primary and acquired resistance to these drugs. Thus, these therapeutic agents could be replaced with more efficient TKI such as Cabozantinib, Axitinib, Lenvatinib and Tivozanib [41-42]. Cabozantinib is an inhibitor of multiple tyrosine kinases including VEGFR, MET and Axl, which are associated with aggressive disease and poor survival [41]. The treatment with Cabozantinib showed an increased PFS in mRCC patients compared with Sunitinib [43]. Axitinib is an anti-angiogenic multi receptor inhibitor that in clinical trials showed greater objective response rates and improved median PFS compared with Sorafenib [4, 42]. Lenvatinib is a multitarget tyrosine kinase inhibitor that inhibits VEGFR, FGFR, PDGFR, RET, and KIT. The administration of this drug has shown antitumor properties in mRCC [44]. Finally, Tivozanib, a novel selective VEGFR inhibitor, was approved for the treatment of advanced RCC by the European Medicines Agency (EMA) [42, 45]. The treatment with Tivozanib showed a PFS advantage compared with Sorafenib, however, overall survival results favored the Sorafenib, but other studies should be carried out to evaluate the efficacy of this drug [45-46]. The combination of Axitinib, Cabozantinib and Lenvatinib with different PD-1 immune checkpoint inhibitors in the first-line setting has shown a superior efficacy in patients with advanced RCC compared with single-drug therapies [6, 47]. Therefore, the latest guidelines recommend the use of these TKI in first-line treatment in combination with immunotherapeutic drugs [6]”.
Precise numbers should be given concerning the efficacy of at least some of new drugs (eg Belzutifan...).
As suggested, we have inserted numbers (%) regarding the efficacy of Belzutifan. Data derive from a phase-II clinical trial.
We have inserted a new paragraph:
“Recently, a phase-II clinical trial by using Belzutifan in patients with renal carcinomas associated with VHL disease was completed. Results indicate that the treatment with Belzutifan induces a reduction in tumor size in most of patients [51]. In particular, about half of enrolled patients (49%) treated with Belzutifan have shown objective response. Only few patients (3%) had progressive disease and one subject (2%) discontinued the treatment because of an adverse event [51]. The efficacy of Belzutifan combined with the modest side effects make this drug an important option for mRCC treatment”.
Moreover, a new reference (51) was added.
Maybe there should be paragraph on new immunotherapies too.
We believe that a paragraph describing immunotherapies should be inserted too. Therefore, we have inserted a short new paragraph briefly describing the main ICIs used for RCC therapy. The paper should not exceed the 4000 word excluding abstract and references. We hope that the Editorial board does not cut this part of paper.
The new paragraph is the following:
“4.6 Immunotherapy
The continuous demand of more efficient therapies for the treatment of mRCC led to the discovery of new agents including immune-oncology drugs such as immune check point inhibitors (ICIs). Immunotherapy for the treatment of advanced RCC by using interleukins or interferon was already attempted, but with poor results [68]. Currently, new immune drugs such as Nivolumab, Ipilimumab, Pembrolizumab and Avelumab were generated. Interestingly, it was observed that the adjuvant treatment with Pembrolizumab, an anti-programmed death 1 (PD-1) monoclonal antibody, prolongs DFS and OS in high risk patients with nonmetastatic RCC subjected to nephrectomy [12]. As previously mentioned, the treatment with ICIs combined with TKIs is recommended for the first line treatment of mRCC [6]. Importantly, the combination of Nivolumab (anti-PD-1 monoclonal antibody) and Ipilimumab (anti-cytotoxic T-lymphocyte-associated protein-4 (CTLA4)) represents the standard of care for RCC patients with intermediate or poor risk [6]. In addition, the dual treatment with the monoclonal antibody anti-PD ligand-1 (PD-L1) Avelumab and the TK inhibitor Axitinib was also approved by FDA for the treatment of all risk groups of RCC [69]. Others innovative immunotherapies include the use of modified cytokines, cellular therapies and anticancer vaccines [70]. Despite promising results obtained with ICI-based therapies, long-term treatments with ICIs in RCC patients may induce mechanisms of resistance decreasing clinical benefits [68]. Therefore, the research of predictive biomarkers for therapeutic response in RCC patient treated with ICIs is needed".
The conclusion should be bullet point 5 and not 4. It should be partly re-written, as I don't think we can consider axitinib or cabozantinib as new drugs.
Point 4 was substituted with 5 in the Conclusions.
We agree that Axitinib and Cabozantinib are not more new TKIs and thus we have partially re-written the Conclusion describing current and possible new drugs for RCC treatment.
“The limited efficacy of therapeutic treatment in mRCC patients is mainly due to the development of drug resistance and severe side effects after first- and second-line treatment. New drugs have been developed and some agents seem very promising since prolong PFS and are well tolerated, while others are ineffective or exhibit high toxicity. Many TKIs in combination with ICIs are currently used in first-line treatment for all risk groups of RCC. Moreover, immuno-oncology therapies are also recommended for intermediate and poor risk of RCC. Other inhibitors have shown interesting anticancer properties; in particular, the HIF2 inhibitor Belzutifan used alone or in combination with other anticancer drugs seem to improve PFS compared with conventional therapies. In addition, the administration of autophagy inhibitor hydroxychloroquine in combination with Everolimus enhances PFS as compared to single treatment with Everolimus in patients suffering for advanced RCC. These agents could represent new therapeutic options for the future treatment of RCC. Conversely, the use of Akt, PI3K and proteasome inhibitors in mRCC patients does not reduce cancer progression and cause severe side effects. MDM2 and TG2 inhibitors as well as ferroptosis activators are able to inhibit tumor cell growth in pre-clinical models of RCC, but future clinical studies are needed to validate their anticancer activities. Furthermore, the treatment with autophagy inhibitors and/or ferroptosis activators could limit drug resistance effects re-sensitizing cancer cell to conventional therapy.
Overall, the research and validation of new targeting drugs combined with the physiopathological features of kidney cancer could lead to personalized medicine improving overall survival and quality of life of mRCC patients.”

Reviewer 2 Report
The purpose of this manuscript is to review new therapeutic interventions for metastatic renal cell carcinoma. The paper describes intracellular signaling pathways and novel agents targeting those pathways.
I found the review of innovative drugs to be thorough, interesting, and well-written. There is, however, a problem with the review of existing therapies, which are currently accepted and recommended.
A major revision should be made mostly but not only to the abstract and introduction to present the most recent systemic therapy practices.
Individual comments are displayed here:
Abstract:
- "Currently, the first line treatment for mRCC patients is the use of antiangiogenic drugs able to inhibit tyrosine kinase receptors (TKI)."
This is not correct. Firs line nowadays are IO+TKI or IO-IO combinations.
Introduction:
The information regarding recommended treatment in first and subsequent lines is not updated and incorrect, for example:
- "In particular, Sunitinib and Sorafenib (VEGFR inhibitors) as well as other anti-angiogenic agents such as Pazopanib, Axitinib and Bevacizumab have been approved by the FDA for the first-line treatment of mRCC [3]."
The information is not full, Cabozantinib was also approved for first line (CABOSUN).
Bevacizumab is not relevant in first line.
- "the current guidelines have not recommended the use of mTOR inhibitors after anti-angiogenic therapy [4]."
This is incorrect, Everolimus in combination with Lenvatinib is recommended after VEGFRi.
2.1 tumor staging
The paper is focused on systemic therapy for mRCC. If you choose to discuss local or locally advanced disease, I recommend discussing adjuvant therapies after resection of a tumor at high risk for recurrence.
- "Stage 2 and 3: RN with a follow-up schedule has baseline abdominal Computed To[1]mography (CT), Magnetic Resonance Imaging (MRI) or ultrasound abdomen and chest CT from 3 to 6 months."
The sentence is unclear and may be irrelevant.
Also, adjuvant treatment with sunitinib (STRAC) or pembrolizumab (KeyNote 564) are recommended for high risk patients as they prolong DFS.
- "Stage 4: Systemic targeted molecular therapies have replaced immunotherapy (e.g. interferons-α) using tyrosine kinase inhibitors (e.g. Axitinib, Sunitinib, Sorafenib and Paz[1]opanib) or rapamycin inhibitors (e.g. Everolimus and Temsirolimus).
Targeted therapies mentioned are no longer relevant. Combination therapy with ipilimumab plus nivolumab or Pembrolizumab+ axitinib/ Pembrolizumab+ Lenvatinib/ Nivolumab+ cabozantinib or cabozantinib alone are recommended for intermediate and poor risk mRCC in first line, and Pembrolizumab+ axitinib/ Pembrolizumab+ Lenvatinib/ Nivolumab+ cabozantinib for favorable risk.
- "Nephrectomy followed by immunotherapy increased survival in patients with metastatic RCC as com[1]pared to immunotherapy or TKIs alone."
What is the reference?
Patients should start systemic treatment as mentioned above. For intermediate risk patients RN is an option in some circumstances.
2.2. Trends in the surgical management of RCC
I don't understand the relevance of this paragraph to the paper.
3.3 p-53 related signaling
" Consistently, we have demonstrated that in different ccRCC cell lines the activation of mTOR promotes the expression of E3 ubiquitin-protein MDM2 that in turn induces p53 ubiquitination and degradation by proteasome [30]"
Who are "we"? the authors names are not as in the reference.
3.4 ferroptosis
"This transcription factor is the predominant Hippo effector in RCC cells and is phosphorylated, retained in the cytosol, and subjected to proteasomal degradation in high cell density conditions [36]."
What is a "hippo effector"?
4. NEW TARGETS FOR RCC TREATMENT
"The current therapies for the treatment of mRCC patients does not improve overall survival"
This is incorrect.
4.1. New TK inhibitors
"Cabozantinib showed an increased PFS in mRCC patients compared with Sunitinib and FDA granted regular approval for this drug in first-line treatment of advanced RCC [4]."
The correct reference is DOI:https://doi.org/10.1016/j.ejca.2018.02.012
"Recently, Axitinib, Cabozantinib and Tivozanib were also approved in combination with different PD-1 immune checkpoint inhibitors such as Pembrolizumab, Nivolumab and Ipilimumab [4, 42]"
This is the standard of care and should be mentioned in the introduction. It is also worth specific part in the discussion.
4.2. HIF2 alpha antagonists and PI3K-related inhibitors
" Belzutifan, is an oral small molecule used for the treatment of solid tumors, including renal cell carcinoma. In phase-1 study, Belzutifan was well tolerated and demonstrated preliminary anti-tumor activity in patients pre-treated with antiangiogenic drugs."
There are already results of phase II in VHL patients and FDA approval: N Engl J Med 2021; 385:2036-2046 DOI: 10.1056/NEJMoa2103425
Author Response
Answer to Reviewer 2
The purpose of this manuscript is to review new therapeutic interventions for metastatic renal cell carcinoma. The paper describes intracellular signaling pathways and novel agents targeting those pathways.
I found the review of innovative drugs to be thorough, interesting, and well-written. There is, however, a problem with the review of existing therapies, which are currently accepted and recommended.
A major revision should be made mostly but not only to the abstract and introduction to present the most recent systemic therapy practices.
Individual comments are displayed here:
- Abstract:
- "Currently, the first line treatment for mRCC patients is the use of antiangiogenic drugs able to inhibit tyrosine kinase receptors (TKI)."
This is not correct. Firs line nowadays are IO+TKI or IO-IO combinations.
As suggested by the Reviewer we have modified the sentence as follows:
“Currently, the first line treatment for mRCC patients is based on antiangiogenic drugs able to inhibit tyrosine kinase receptors (TKI) in combination with immuno-oncology (IO) therapy or IO-IO treatments”.
- Introduction:
The information regarding recommended treatment in first and subsequent lines is not updated and incorrect, for example:
- "In particular, Sunitinib and Sorafenib (VEGFR inhibitors) as well as other anti-angiogenic agents such as Pazopanib, Axitinib and Bevacizumab have been approved by the FDA for the first-line treatment of mRCC [3]."
The information is not full. Cabozantinib was also approved for first line (CABOSUN). Bevacizumab is not relevant in first line.
We have substituted Bevacizumab with Cabozantinib. The new sentence is:
“In particular, Sunitinib and Sorafenib (VEGFR inhibitors) as well as other anti-angiogenic agents such as Pazopanib, Axitinib and Cabozantinib have been approved by the FDA for the first-line treatment of mRCC [3]”.
- "the current guidelines have not recommended the use of mTOR inhibitors after anti-angiogenic therapy [4]."
This is incorrect, Everolimus in combination with Lenvatinib is recommended after VEGFRi.
The sentence above was modified adding the word “alone”. The new sentence is the following:
“….therefore, the current guidelines have not recommended the use of mTOR inhibitors alone after anti-angiogenic therapy [4]”.
2.1 tumor staging
The paper is focused on systemic therapy for mRCC. If you choose to discuss local or locally advanced disease, I recommend discussing adjuvant therapies after resection of a tumor at high risk for recurrence.
- "Stage 2 and 3: RN with a follow-up schedule has baseline abdominal Computed Tomography [1] (CT), Magnetic Resonance Imaging (MRI) or ultrasound abdomen and chest CT from 3 to 6 months."
The sentence is unclear and may be irrelevant.
Also, adjuvant treatment with sunitinib (STRAC) or pembrolizumab (KeyNote 564) are recommended for high risk patients as they prolong DFS.
- "Stage 4: Systemic targeted molecular therapies have replaced immunotherapy (e.g. interferons-α) using tyrosine kinase inhibitors (e.g. Axitinib, Sunitinib, Sorafenib and Pazopanib) or rapamycin inhibitors (e.g. Everolimus and Temsirolimus).
Targeted therapies mentioned are no longer relevant. Combination therapy with ipilimumab plus nivolumab or Pembrolizumab+ axitinib/ Pembrolizumab+ Lenvatinib/ Nivolumab+ cabozantinib or cabozantinib alone are recommended for intermediate and poor risk mRCC in first line, and Pembrolizumab+ axitinib/ Pembrolizumab+ Lenvatinib/ Nivolumab+ cabozantinib for favorable risk.
- "Nephrectomy followed by immunotherapy increased survival in patients with metastatic RCC as compared to immunotherapy or TKIs alone."
What is the reference?
Patients should start systemic treatment as mentioned above. For intermediate risk patients RN is an option in some circumstances.
We have re-written the paragraph 2.1 as suggested by the Reviewer. Moreover, two new references have been inserted (new references 6 and 12). The new paragraph is the following:
“2.1. Tumor staging and patient management
After diagnosis, RCC patients are risk stratified into favourable, intermediate, and poor risk categories. Recently, the integration of molecular data with annotated genomic models showed improved stratification of patients across risk groups, although they have not yet been usually incorporated into clinical practice [7, 9-10]. New therapies are developing based on this discovery and new concepts regarding disease management to prevent overtreatment or toxicity are being studied. The therapeutic options and management of RCC are stage-dependent; consequently, accurate staging is essential to effective management [9]. Kidney carcinomas can be classified in the following groups:
Stage 1a: Tumors confined to the kidney. Complete surgical resection of the tumor with nephron-sparing surgery (NSS) is recommended.
Stage 1b: Partial or radical nephrectomy (PN and RN, respectively) has similar results.
Stage 2: Tumor confined to the organ with a size up to 10 cm. Surgery, including PN and RN, is an option for the resection of tumor masses.
Stage 3: Tumor invades renal sinus, pelvicalyceal system and vena cava. PN and RN are recommended as surgical treatment.
Stage 4: Tumor invades beyond Gerota fascia and adrenal gland. The surgical treatment for these patients is debated. In fact, different studies suggest that nephrectomy may improve OS in mRCC subjects treated with targeted therapy. Conversely, the CARMENA study shows that OS and PFS in mRCC patients treated with Sunitinib alone was not inferior to nephrectomy followed by Sunitinib administration [11].
Metastatic RCC patients are now treated with systemic targeted molecular therapies that have replaced immunotherapy (e.g. interferons-α). In particular, the standard care for the treatment of favourable risk patients is the use of Pembrolizumab/Axitinib, Pembrolizumab/Lenvatinib or Nivolumab/Cabozantinib. For intermediate and poor risk patients is recommended the treatment with Nivolumab/Cabozantinib, Pembrolizumab/Axitinib, Pembrolizumab/Lenvatinib or Nivolumab/Ipilimumab [6].
High risk nonmetastatic RCC patients subjected to nephrectomy might be treated with Pembrolizumab, because this treatment improves both disease free survival (DFS) and OS and showed lower toxicity compared with TKIs [12]. These findings support Pembrolizumab as new standard of care for nephrectomy RCC patients with high risk to develop metastasis [12].
2.2. Trends in the surgical management of RCC
I don't understand the relevance of this paragraph to the paper.
This paper is mainly focused on the future pharmacological treatments of mRCC patients. However, we believe that some notions on the surgical techniques used for tumor resection might give further information and improve the paper clinically. If the Reviewer considers it irrelevant, we can delete it.
3.3 p-53 related signaling
"Consistently, we have demonstrated that in different ccRCC cell lines the activation of mTOR promotes the expression of E3 ubiquitin-protein MDM2 that in turn induces p53 ubiquitination and degradation by proteasome [30]"
Who are "we"? authors names are not as in the reference.
The reference (old number 30 and now 33) is a work published by us.
This is the reference:
Mangolini A, Bonon A, Volinia S, Lanza G, Gambari R, Pinton P, Russo GR, Del Senno L, Dell'Atti L, Aguiari G. (2014). Differential expression of microRNA501-5p affects the aggressiveness of clear cell renal carcinoma. FEBS Open Bio. 4:952-65. doi: 10.1016/j.fob.2014.10.016.
Anyway, we have changed “we have demonstrated..” with “it was reported that…”
3.4 ferroptosis
"This transcription factor is the predominant Hippo effector in RCC cells and is phosphorylated, retained in the cytosol, and subjected to proteasomal degradation in high cell density conditions [36]."
What is a "hippo effector"?
In animals, the Hippo pathway regulates organ size through the modulation of cell proliferation and apoptosis. This pathway takes its name from the protein kinase Hippo, a component like TAZ of this signaling pathway. Mutations in the gene codifying for this kinase lead to tissue overgrowth, showing a hippopotamus-like phenotype.
- NEW TARGETS FOR RCC TREATMENT
"The current therapies for the treatment of mRCC patients does not improve overall survival"
This is incorrect.
We have modified the sentence as follows:
“The pharmacological therapies for the treatment of mRCC patients move fast. In fact, the monotherapy with tyrosine kinase inhibitors targeting the VEGF receptor to inhibit angiogenesis was replaced with more effective therapies.
Here, we explore current and novel possible targeted drug for the treatment of mRCC.”
Moreover, the paragraph 4.1 was deeply modified inserting the correct data
4.1. New TK inhibitors
"Cabozantinib showed an increased PFS in mRCC patients compared with Sunitinib and FDA granted regular approval for this drug in first-line treatment of advanced RCC [4]."
The correct reference is DOI: https://doi.org/10.1016/j.ejca.2018.02.012
As suggested, we have inserted the new reference (number 43).
"Recently, Axitinib, Cabozantinib and Tivozanib were also approved in combination with different PD-1 immune checkpoint inhibitors such as Pembrolizumab, Nivolumab and Ipilimumab [4, 43]"
This is the standard of care and should be mentioned in the introduction. It is also worth specific part in the discussion.
We have added in Introduction section a sentence mentioning the standard of care:
“Currently, combined therapies with TKIs and immune checkpoint inhibitors (ICI) or combinations of ICIs are used for the first-line treatment of RCC [6]. However,”
Moreover, we have inserted in the 4.1 paragraph the following sentence:
“The combination of Axitinib, Cabozantinib and Lenvatinib with different PD-1 immune checkpoint inhibitors in the first-line setting has shown a superior efficacy in patients with advanced RCC compared with single-drug therapies [6, 47]. Therefore, the latest guidelines recommend the use of these TKI in first-line treatment in combination with immunotherapeutic drugs [6].”
4.2. HIF2 alpha antagonists and PI3K-related inhibitors
"Belzutifan, is an oral small molecule used for the treatment of solid tumors, including renal cell carcinoma. In phase-1 study, Belzutifan was well tolerated and demonstrated preliminary anti-tumor activity in patients pre-treated with antiangiogenic drugs."
There are already results of phase II in VHL patients and FDA approval: N Engl J Med 2021; 385:2036-2046 DOI: 10.1056/NEJMoa2103425
We have inserted a new paragraph containing these updates:
“Recently, a phase-II clinical trial by using Belzutifan in patients with renal carcinomas associated with VHL disease was completed. Results indicate that the treatment with Belzutifan induces a reduction in tumor size in most of patients [51]. In particular, about half of enrolled patients (49%) treated with Belzutifan have shown objective response. Only few patients (3%) had progressive disease and one subject (2%) discontinued the treatment because of an adverse event [51]. The efficacy of Belzutifan combined with the modest side effects make this drug an important option for mRCC treatment”.
Moreover, a new reference (51) was added.

Round 2
Reviewer 1 Report
In the paragraph 2.1:
Management of localized disease and metastatic disease must be separated
Favorable, intermediate and poor risk refer to metastatic disease
Author Response
Answers to Reviewer 1 round-2
Comments and Suggestions for Authors
In the paragraph 2.1:
Management of localized disease and metastatic disease must be separated
Favorable, intermediate and poor risk refer to metastatic disease
As suggested, we have modified the paragraph 2.1. We have separated tumor staging and patient treatment. We have added a new paragraph (2.3) with the management of RCC patients post-nephrectomy. The new paragraph is the following:
2.3. Management of RCC patients after diagnosis
Nonmetastatic RCC patients with localized disease and subjected to partial or radical nephrectomy are monitored by surveillance. Follow-up should be personalized based on patient needs. Adjuvant therapy by using TKIs for high risk nonmetastatic RCC patients after nephrectomy was approved by FDA, but is not recommended by EAU guidelines. In fact, the S-TRAC study showed that the treatment with Sunitinib improved disease free survival (DFS), but without any OS benefit [10]. Recently, the availability of novel drugs including ICIs has improved the efficacy of adjuvant therapies. In this regard, the Keynote-564 trial reported that the treatment with Pembrolizumab enhanced DFS in high risk nonmetastatic RCC patients subjected to nephrectomy as compared to placebo [10, 18]. These observations indicate that Pembrolizumab could be used as adjuvant therapy for high risck RCC. However, before recommending this therapy it would be appropriate to know data on OS, currently, not yet available [10].
Metastatic RCC patients are risk stratified into favourable, intermediate, and poor risk categories. Recently, the integration of molecular data with annotated genomic models showed improved stratification of patients across risk groups. In particular, the standard of care for the treatment of favourable risk patients is the use of Pembrolizumab/Axitinib, Pembrolizumab/Lenvatinib or Nivolumab/Cabozantinib. For intermediate and poor risk is recommended the treatment with Nivolumab/Cabozantinib, Pembrolizumab/Axitinib, Pembrolizumab/Lenvatinib or Nivolumab/Ipilimumab [6].

Reviewer 2 Report
- Despite the slight changes, the introduction section does not yet accurately describe the current practice.
- In section 2: CLINICAL ASPECTS, there are errors that indicate a lack of familiarity with clinical practice. The description of the disease stage is inaccurate, the discussion of post-nephrectomy risk categorization for local disease is incorrect (favorable, intermediate and poor are used in metastatic disease) and the discussion of CARMENA trial instead of trials of adjuvant therapy as I have suggested, is irrelevant. The authors mentioned adjuvant pembrolizumab but however, they misunderstood the results and the paragraph is incorrect.
- Section 4.1 was revised properly
- The new section 4.6 of immunotherapy is good. Only minor changes are needed:
"Interestingly, it was observed that the adjuvant treatment with Pembrolizumab, an anti-programmed death 1 (PD-1) monoclonal antibody, prolongs DFS and OS in high risk patients with nonmetastatic RCC subjected to nephrectomy [12]."
OS is not prolonged, date immature.
"In addition, the dual treatment with the monoclonal antibody anti-PD ligand-1 (PD-L1) Avelumab and the TK inhibitor Axitinib was also approved by FDA for the treatment of all risk groups of RCC [69]."
FDA approved also pembro-axi, nivo-cabo and pembro-lenva, with OS benefit over sunitinib.
Author Response
Answer to Reviewer 2 (Round-2)
Comments and Suggestions for Authors
- Despite the slight changes, the introduction section does not yet accurately describe the current practice.
- In section 2: CLINICAL ASPECTS, there are errors that indicate a lack of familiarity with clinical practice. The description of the disease stage is inaccurate, the discussion of post-nephrectomy risk categorization for local disease is incorrect (favorable, intermediate and poor are used in metastatic disease) and the discussion of CARMENA trial instead of trials of adjuvant therapy as I have suggested, is irrelevant.
The authors mentioned adjuvant pembrolizumab but however, they misunderstood the results and the paragraph is incorrect.
As suggested by the reviewer, we have re-written the chapter 2.1 as follows:
2.1. Tumor staging
The therapeutic options and management of RCC are stage-dependent; consequently, accurate staging is essential to effective management [9]. Kidney primary tumors can be classified in the following groups:
T1: Tumor is confined to the kidney and is subdivided in T1a and T1b. T1a: tumor is 4 cm or less in size; T1b: tumor is between 4 and 7 cm.
T2: Tumor is limited to the organ and is divided in T2a and T2b. T2a: tumor is between 7 and 10 cm; T2b: tumor is more than 10 cm.
T3: Tumor extends into major veins or perinephric tissues and is divided in T3a, T3b and T3c. T3a: tumor extends into the renal vein or invades the pelvicalyceal system or spreads into perirenal and/or renal sinus fat; T3b: tumor extends into vena cava below diaphragm; T3c: tumor extends into the vena cava above the diaphragm or invades the wall of the vena cava.
T4: Tumor invades beyond Gerota fascia (including contiguous extension into the ipsilateral adrenal gland).
Renal cancer staging is based on Tumor Node Metastasis (TNM) classification system and includes four different stages.
Stage I: Includes T1 tumors without lymph node or distance metastasis. Partial Nephrectomy (PN) or Radical Nephrectomy (RN) are recommended.
Stage II: Includes T2 tumors without lymph node or distance metastasis. Surgery including PN and RN is an option for the resection of tumor masses.
Stage III: Includes T3 tumors without lymph node or distance metastasis and T1-T3 tumors with metastasis in regional lymph nodes, but not distance metastasis. RN or PN (if clinically indicated) are recommended as surgical treatment.
Stage IV: Includes T4 tumors with any lymph node without distance metastasis as well as T1-T4 tumors with any lymph node and distance metastasis. RN could be an option for patients with localized disease. Cytoreductive nephrectomy (CN) before systemic therapy might be an option for mRCC patients with surgically treatable primary tumors. In fact, studies conducted in the cytokine era showed that the combined treatment with CN and interferon enhanced survival of patients compared with subjects treated with interferon alone. Conversely, the CARMENA study reported that OS in patients treated with Sunitinib alone was not inferior to CN followed by Sunitinib [10-11]. Based on these observations, immediate systemic treatment for mRCC patients is recommended [10].
Moreover, we have inserted a new chapter 2.3 with patient management post-nephrectomy. The new chapter is the following:
2.3. Management of RCC patients after diagnosis
Nonmetastatic RCC patients with localized disease and subjected to partial or radical nephrectomy are monitored by surveillance. Follow-up should be personalized based on patient needs. Adjuvant therapy by using TKIs for high risk nonmetastatic RCC patients after nephrectomy was approved by FDA, but is not recommended by EAU guidelines. In fact, the S-TRAC study showed that the treatment with Sunitinib improved disease free survival (DFS), but without any OS benefit [10]. Recently, the availability of novel drugs including ICIs has improved the efficacy of adjuvant therapies. In this regard, the Keynote-564 trial reported that the treatment with Pembrolizumab enhanced DFS in high risk nonmetastatic RCC patients subjected to nephrectomy as compared to placebo [10, 18]. These observations indicate that Pembrolizumab could be used as adjuvant therapy for high risck RCC. However, before recommending this therapy it would be appropriate to know data on OS, currently, not yet available [10].
Metastatic RCC patients are risk stratified into favourable, intermediate, and poor risk categories. Recently, the integration of molecular data with annotated genomic models showed improved stratification of patients across risk groups. In particular, the standard of care for the treatment of favourable risk patients is the use of Pembrolizumab/Axitinib, Pembrolizumab/Lenvatinib or Nivolumab/Cabozantinib. For intermediate and poor risk is recommended the treatment with Nivolumab/Cabozantinib, Pembrolizumab/Axitinib, Pembrolizumab/Lenvatinib or Nivolumab/Ipilimumab [6].
- Section 4.1 was revised properly
- The new section 4.6 of immunotherapy is good. Only minor changes are needed:
"Interestingly, it was observed that the adjuvant treatment with Pembrolizumab, an anti-programmed death 1 (PD-1) monoclonal antibody, prolongs DFS and OS in high risk patients with nonmetastatic RCC subjected to nephrectomy [12]."
OS is not prolonged, date immature.
As suggested, we have deleted “and OS” in the sentence above (section 4.6 of the paper).
"In addition, the dual treatment with the monoclonal antibody anti-PD ligand-1 (PD-L1) Avelumab and the TK inhibitor Axitinib was also approved by FDA for the treatment of all risk groups of RCC [69]."
FDA approved also pembro-axi, nivo-cabo and pembro-lenva, with OS benefit over sunitinib.
Of course.
We have previously mentioned more times the standard of care by using pembro-axi, nivo-cabo and pembro-lenva and we did not want to repeat it again.
Anyway, the phrase:” In addition, the dual treatment with the monoclonal antibody anti-PD ligand-1 (PD-L1) Avelumab and the TK inhibitor Axitinib was also approved by FDA for the treatment of all risk groups of RCC [70].” was moved above.
The new sentences of chapter 4.6 becomes the following:
“As previously mentioned, the treatment with ICIs combined with TKIs is recommended for the first line treatment of mRCC [6]. In addition, the dual treatment with the monoclonal antibody anti-PD ligand-1 (PD-L1) Avelumab and the TK inhibitor Axitinib was also approved by FDA for the treatment of all risk groups of RCC [70].”

Round 3
Reviewer 2 Report
I think the revision have made your article more up to date and therefore more relevant. Good Luck